# Multiple pathways of toxicity induced by *C9orf72* dipeptide repeat aggregates and G$_4$C$_2$ RNA in a cellular model

Frédéric Frottin[1,2]*, Manuela Pérez-Berlanga[1,3], F Ulrich Hartl[1]*, Mark S Hipp[1,4,5]*

[1]Max Planck Institute of Biochemistry, Martinsried, Germany; [2]Université Paris-Saclay, CEA, CNRS, Institute for Integrative Biology of the Cell (I2BC), Gif-sur-Yvette, France; [3]Department of Quantitative Biomedicine, University of Zurich, Zurich, Switzerland; [4]Department of Biomedical Sciences of Cells and Systems, University Medical Center Groningen, University of Groningen, Groningen, Netherlands; [5]School of Medicine and Health Sciences, Carl von Ossietzky University Oldenburg, Oldenburg, Germany

**Abstract** The most frequent genetic cause of amyotrophic lateral sclerosis and frontotemporal dementia is a G$_4$C$_2$ repeat expansion in the *C9orf72* gene. This expansion gives rise to translation of aggregating dipeptide repeat (DPR) proteins, including poly-GA as the most abundant species. However, gain of toxic function effects have been attributed to either the DPRs or the pathological G$_4$C$_2$ RNA. Here, we analyzed in a cellular model the relative toxicity of DPRs and RNA. Cytoplasmic poly-GA aggregates, generated in the absence of G$_4$C$_2$ RNA, interfered with nucleocytoplasmic protein transport, but had little effect on cell viability. In contrast, nuclear poly-GA was more toxic, impairing nucleolar protein quality control and protein biosynthesis. Production of the G$_4$C$_2$ RNA strongly reduced viability independent of DPR translation and caused pronounced inhibition of nuclear mRNA export and protein biogenesis. Thus, while the toxic effects of G$_4$C$_2$ RNA predominate in the cellular model used, DPRs exert additive effects that may contribute to pathology.

**\*For correspondence:**
frederic.frottin@i2bc.paris-saclay.fr (FF);
uhartl@biochem.mpg.de (FUH);
m.s.hipp@umcg.nl (MSH)

## Introduction

Expansion of a GGGGCC hexanucleotide repeat (hereafter G$_4$C$_2$) within the first intron of the *C9orf72* gene is the most frequent genetic cause of amyotrophic lateral sclerosis (ALS) and fronto-temporal dementia (FTD) (*DeJesus-Hernandez et al., 2011*; *Renton et al., 2011*). Mutant *C9orf72* in patients suffering from ALS/FTD can have more than a thousand G$_4$C$_2$ repeats, while healthy individuals possess usually less than 20 repeats (*Gijselinck et al., 2016*; *Nordin et al., 2015*). Transcripts with expanded G$_4$C$_2$ tracts are translated by repeat associated non-AUG (RAN) translation in all reading frames and in both strands, resulting in the synthesis of five different dipeptide repeat proteins (DPRs): poly-GA, poly-GR, poly-GP, poly-PR, and poly-PA (*Ash et al., 2013*; *Gendron et al., 2013*; *Mackenzie et al., 2015*; *Mori et al., 2013a*; *Mori et al., 2013c*; *Zu et al., 2013*), all of which have been detected in patient brains (*Mori et al., 2013a*; *Mori et al., 2013c*; *Zu et al., 2013*). Poly-GA is the most abundant of the DPRs, followed by the other sense strand-encoded forms (poly-GP and poly-GR) (*Mackenzie et al., 2015*; *Mori et al., 2013c*; *Schludi et al., 2015*). In patient brain and cellular models, DPRs accumulate in deposits that can be found in the nucleus and cytoplasm, including neurites (*Ash et al., 2013*; *Gendron et al., 2013*; *Mackenzie et al., 2015*; *Mori et al., 2013a*; *Mori et al., 2013c*; *Schludi et al., 2015*; *Zu et al., 2013*). Poly-GA aggregates are localized mainly in the cytoplasm (*Davidson et al., 2016*; *Lee et al., 2017*; *Mackenzie et al., 2015*; *Zhang et al., 2016*), whereas arginine-containing DPRs (R-DPRs; poly-GR and poly-PR) accumulate in the nucleus

(*Mackenzie et al., 2015*; *Schludi et al., 2015*). R-DPRs have also been shown in cellular models to localize to the nucleolus (*Kwon et al., 2014*; *Lee et al., 2017*; *May et al., 2014*; *Moens et al., 2019*; *Wen et al., 2014*; *White et al., 2019*; *Yamakawa et al., 2015*; *Zhang et al., 2014*). However, in patients, poly-GR and poly-PR predominantly form cytoplasmic deposits, with only a fraction of cells containing para-nucleolar inclusions that co-localize with silent DNA (*Mackenzie et al., 2015*; *Schludi et al., 2015*). Interestingly, the less frequent intranuclear poly-GA inclusions in both cell models and patient brain are excluded from the nucleoli (*Schludi et al., 2015*).

Both loss- and gain-of-function mechanisms have been suggested to contribute to *C9orf72* pathology (reviewed in *Balendra and Isaacs, 2018*; *Jiang and Ravits, 2019*; *Swinnen et al., 2018*). Despite its location in a non-coding part of the gene, the $G_4C_2$ expansion can alter the expression level of the C9ORF72 protein (*Rizzu et al., 2016*; *Shi et al., 2018*; *Waite et al., 2014*). However, *C9orf72* knockout mouse models failed to fully recapitulate ALS- or FTD-related neurodegenerative phenotypes, suggesting that loss of C9ORF72 protein is not the only contributor to pathology (*Atanasio et al., 2016*; *Burberry et al., 2016*; *Burberry et al., 2020*; *Jiang et al., 2016*; *Koppers et al., 2015*; *Lagier-Tourenne et al., 2013*; *O'Rourke et al., 2016*; *Panda et al., 2013*; *Sudria-Lopez et al., 2016*; *Sullivan et al., 2016*; *Suzuki et al., 2013*; *Ugolino et al., 2016*; *Zhu et al., 2020*).

Toxic functions induced by the $G_4C_2$ expansion have been studied in various cellular and animal models, and both RNA- and protein-based mechanisms of toxicity have been proposed (*Arzberger et al., 2018*). However, the main contributor to gain of toxic function in the disease remains to be defined. Pathological $G_4C_2$ mRNA forms stable G-quadruplexes in the nucleus that retain RNA binding proteins and induce splicing defects (*Cooper-Knock et al., 2014*; *Donnelly et al., 2013*; *Gitler and Tsuiji, 2016*; *Haeusler et al., 2014*; *Sareen et al., 2013*; *Simón-Sánchez et al., 2012*; *Xu et al., 2013*), but has also been observed in the cytoplasm (*Cooper-Knock et al., 2015*; *Liu et al., 2016*; *Mizielinska et al., 2017*; *Ohki et al., 2017*; *Swinnen et al., 2018*). $G_4C_2$-containing RNA and associated DPRs have been reported to alter the nucleocytoplasmic transport machinery (*Boeynaems et al., 2016*; *Freibaum et al., 2015*; *Jovičić et al., 2015*; *Zhang et al., 2015*). Moreover, R-DPRs can interact with membrane-free phase-separated compartments, such as the nucleolus, causing nucleolar stress and dysfunction of nucleolar quality control, impairment of nucleocytoplasmic trafficking and protein translation, as well as induction of stress granule formation (*Frottin et al., 2019*; *Hayes et al., 2020*; *Kanekura et al., 2016*; *Kwon et al., 2014*; *Lee et al., 2016*; *Mizielinska et al., 2017*; *Moens et al., 2019*; *Radwan et al., 2020*; *Shi et al., 2017*; *Tao et al., 2015*; *Vanneste et al., 2019*; *White et al., 2019*; *Yamakawa et al., 2015*; *Zhang et al., 2018b*). Additionally, cytoplasmic poly-GA aggregates associate extensively with proteasomes and other components of the ubiquitin proteasome system, and interfere with proteasome activity (*Al-Sarraj et al., 2011*; *Guo et al., 2018*; *Khosravi et al., 2020*; *May et al., 2014*; *Mori et al., 2013c*; *Riemslagh et al., 2019*), as well as induce mislocalization of nuclear proteins (*Khosravi et al., 2017*; *Nihei et al., 2020*; *Nonaka et al., 2018*; *Solomon et al., 2018*). In mice expressing poly-GA, the aggregates have also been observed to sequester nuclear pore components (*Zhang et al., 2016*), thereby interfering with nucleocytoplasmic protein transport. In light of these studies, it seems likely that multiple toxicity mechanisms contribute to pathogenesis (*Balendra and Isaacs, 2018*).

Here, we used $G_4C_2$ repeat-containing constructs as well as synthetic DPR constructs not generating $G_4C_2$ RNA to analyze the relative contributions of RNA and DPR species to pathology in a cellular model. The mRNA and protein constructs analyzed were of equivalent length. By retaining DPR proteins in the cytoplasm or targeting them to the nucleus, we found that cytoplasmic, but not nuclear poly-GA, aggregates impaired nucleocytoplasmic transport. However, a direct comparison of length-matched constructs showed that nuclear poly-GA was more toxic and interfered with nucleolar protein quality control and protein synthesis. Importantly, by leveraging the finding that cytoplasmic poly-GA does not induce toxicity in the cellular model system, we were able to isolate the contribution of the $G_4C_2$ repeat RNA to cellular pathology. The $G_4C_2$-containing RNA species induced a strong accumulation of poly-adenylated mRNA within the nucleus and dramatically inhibited protein biosynthesis. Thus, poly-GA protein and $G_4C_2$ RNA interfere with multiple key cellular pathways, with the RNA component exerting the major toxic effects limiting cell viability.

# Results

## Nuclear and cytoplasmic poly-GA aggregates differ in toxicity

Both $G_4C_2$ RNA and DPR proteins resulting from the pathological *C9orf72* expansion have been suggested to mediate gain of toxic function effects in various models (reviewed in *Balendra and Isaacs, 2018*). To investigate the contribution of poly-GA proteins to cellular pathology, we engineered ATG-driven synthetic constructs expressing 65 GA repeats fused C-terminally to GFP ($GA_{65}$-GFP). Notably, these constructs do not contain any $G_4C_2$ hexanucleotide motifs (*Figure 1A, Figure 1—figure supplement 1A*).

Expression of $GA_{65}$-GFP in HEK293 cells resulted in the formation of bright fluorescent inclusions in most transfected cells. The inclusions were generally cytoplasmic, except for a small fraction of cells with nuclear foci (*Figure 1B*). This phenotype is consistent with reports on the localization of poly-GA aggregates in patient brain, where poly-GA aggregates are also observed in the neuronal

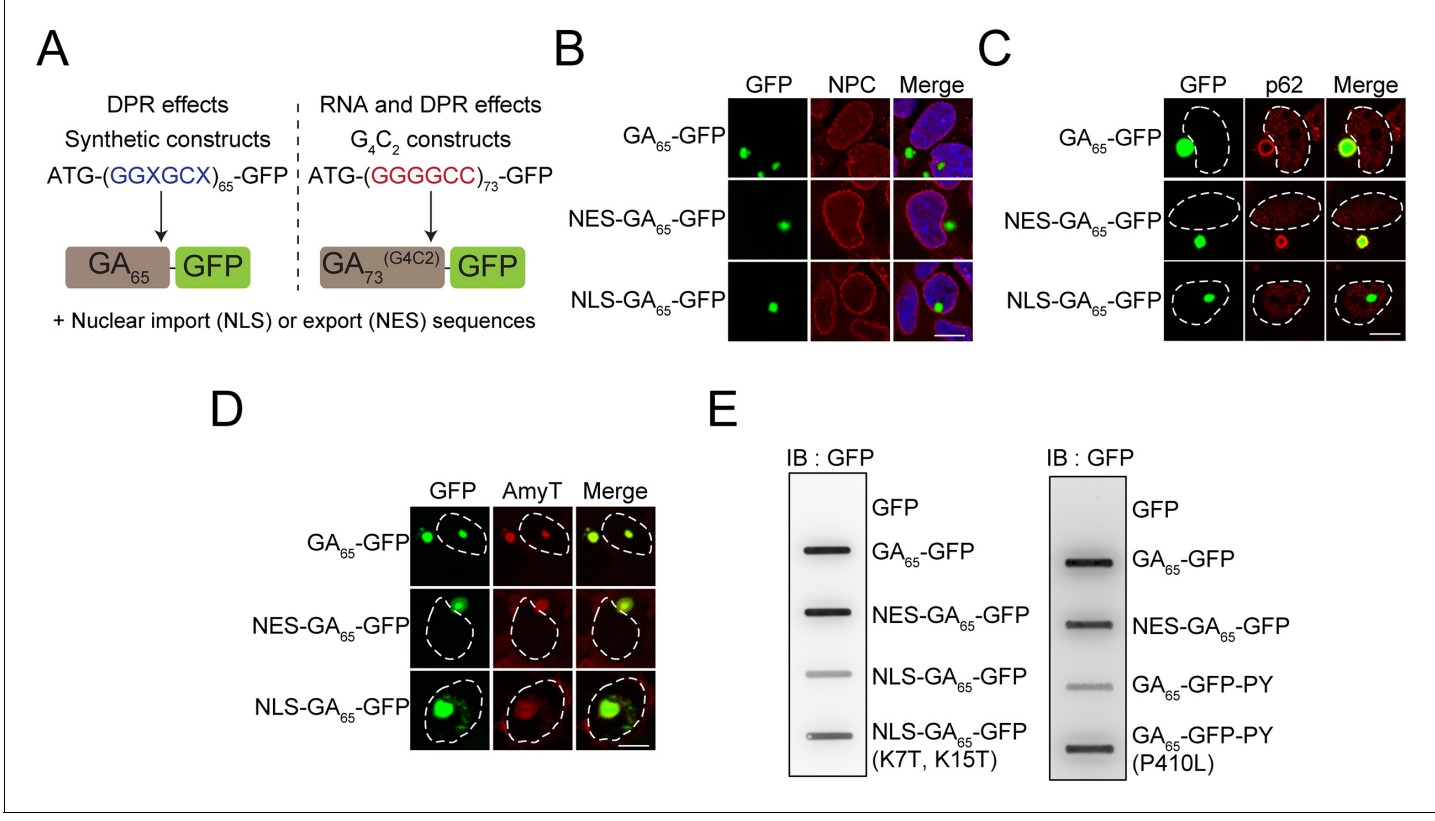

**Figure 1.** Cytoplasmic and nuclear poly-GA aggregates are amyloid-like but differ in solubility. (**A**) Schematic representation of the poly-GA encoding constructs used. Two different classes of poly-GA encoding constructs were used to study dipeptide repeat protein (DPR)-mediated toxicity in the presence or absence of RNA repeat regions: synthetic constructs that do not contain $G_4C_2$ motifs and are translated into $GA_{65}$ (left), and constructs containing 73 $G_4C_2$ repeats that encode $GA_{73}$ (right). Both classes contained ATG start codons and were fused in frame to GFP. N-terminal nuclear import (NLS) or export sequences (NES) were present when indicated. (**B**) The indicated constructs were transfected into HEK293 cells. Antibodies against nuclear pore complexes (NPC, red) were used to detect the nuclear membrane, poly-GA was visualized by GFP fluorescence (green), and nuclei were counterstained with DAPI. (**C**) NES-$GA_{65}$-GFP aggregates co-localize with p62. The indicated constructs were transfected into HEK293 cells. p62 (red) was detected by immunofluorescence, and poly-GA was visualized by GFP fluorescence (green). White dashed lines delineate the nucleus based on DAPI staining. (**D**) Cytoplasmic and nuclear aggregates can be stained with an amyloid-specific dye. The indicated constructs were transfected into HEK293 cells and stained with AmyTracker (AmyT, red). Poly-GA DPRs were visualized by GFP fluorescence (green). White dashed lines delineate the nucleus based on DAPI staining. (**E**) Cytoplasmic GA aggregates are SDS insoluble. The indicated constructs were transfected into HEK293 cells. GFP was expressed as a soluble control protein. Cells were lysed and analyzed for SDS-insoluble poly-GA aggregates by filter retardation assay. GFP antibody was used for detection. Scale bars represent 10 µm.

The online version of this article includes the following figure supplement(s) for figure 1:

**Figure supplement 1.** Cytoplasmic and nuclear poly-GA aggregates differ in their staining properties.

cytoplasm and nucleus (*Mori et al., 2013c*; *Schludi et al., 2015*). Using engineered β-sheet proteins, we have previously reported that otherwise identical aggregation-prone proteins display distinct toxic properties when targeted to different cellular compartments (*Frottin et al., 2019*; *Vincenz-Donnelly et al., 2018*; *Woerner et al., 2016*). To test whether this is also the case for poly-GA, we generated compartment-specific variants of the poly-GA proteins. We restricted the expression of poly-GA to the cytoplasm by adding a nuclear export signal (NES-GA$_{65}$-GFP) or targeted the protein to the nucleus using a double SV40 nuclear localization signal (NLS-GA$_{65}$-GFP). NES-GA$_{65}$-GFP accumulated in the cytoplasm and formed inclusions similar to those of GA$_{65}$-GFP (*Figure 1B*). Directing the protein to the nucleus resulted in an increased number of cells with nuclear aggregates (*Figure 1B*). However, a number of NLS-GA$_{65}$-GFP-expressing cells also contained cytoplasmic inclusions, suggesting that aggregate formation in these cells occurred before the transport of the poly-GA proteins into the nucleus.

Poly-GA forms p62-positive inclusions in the cytoplasm of neurons (*Guo et al., 2018*). We were able to replicate this phenotype in our cellular system and observed p62-positive poly-GA inclusions of GA$_{65}$-GFP and NES-GA$_{65}$-GFP (*Figure 1C*; *May et al., 2014*; *Mori et al., 2013c*). Both nuclear and cytoplasmic poly-GA inclusions were stained throughout with AmyT, a small amyloid-specific dye (*Figure 1D*). The aggregates were also recognized by the anti-amyloid antibody (OC) (*Figure 1—figure supplement 1B*), which recognizes generic epitopes common to amyloid fibrils and fibrillar oligomers (*Kayed et al., 2007*). However, while the nuclear aggregates stained homogeneously with OC, cytoplasmic GA$_{65}$-GFP inclusions stained less well and rather showed only a peripheral reaction with the dye (*Figure 1—figure supplement 1B*). The differential antibody accessibility of the inclusion core suggests that the nuclear and cytoplasmic DPR aggregates, though both amyloid-like, differ in structural properties such as packing density.

Analysis of the solubility of the nuclear and cytoplasmic GA$_{65}$-GFP aggregates supported this interpretation. NES-GA$_{65}$-GFP was retained in a filter retardation assay in the presence of SDS, while most NLS-GA$_{65}$-GFP passed through the filter (*Figure 1E*). The difference in detergent solubility was not due to the presence of the NLS sequence since similar results were obtained using the FUS-derived C-terminal nuclear localization signal (PY) (*Gal et al., 2011*; *Figure 1E*). Moreover, disabling of the nuclear targeting sequences by point mutations resulted in the reappearance of SDS-insoluble aggregates (*Figure 1E*). Thus, despite being amyloid-like, nuclear and cytoplasmic poly-GA aggregates have different physico-chemical properties.

The granular component (GC) of the nucleolus has recently been shown to function as a protein quality control compartment (*Frottin et al., 2019*; *Woerner et al., 2016*). Interestingly, the nuclear aggregates of NLS-GA$_{65}$-GFP altered the localization of the GC marker protein nucleophosmin (NPM1), while cytosolic NES-GA$_{65}$-GFP had no such effect (*Figure 2A*). Nuclear GA$_{65}$-GFP aggregates induced the dislocation of NPM1 from the GC phase of the nucleolus, with accumulation of NPM1 at the aggregate periphery (*Figure 2A*). However, the nuclear GA$_{65}$-GFP deposits did not alter the distribution of the RNA polymerase I subunit RPA40, a marker of the fibrillar center of the nucleolus, the site of rRNA synthesis (*Figure 2B*). The NPM1-containing GC phase is responsible for pre-ribosome particle assembly and also accommodates proteins that have misfolded upon stress. These proteins enter the GC phase and are maintained in a state competent for refolding and repartitioning to the nucleoplasm upon recovery from stress (*Frottin et al., 2019*). In contrast to poly-GA, R-DPRs enter the GC phase of the nucleolus and convert it from liquid-like to a more hardened state, thereby impairing its quality control function (*Frottin et al., 2019*; *Lee et al., 2016*). To test whether nuclear poly-GA aggregates also affect nucleolar quality control, we expressed the synthetic poly-GA constructs in HEK293 cells together with nuclear firefly luciferase fused to the red fluorescent protein mScarlet (NLS-LS), a metastable protein that enters the GC phase upon stress-induced misfolding (*Frottin et al., 2019*). In control cells, NLS-LS accumulated in the nucleolus upon heat stress and largely repartitioned to the nucleoplasm within 2 hr of recovery (*Figure 2C*, *Figure 2—figure supplement 1A*). A similar result was obtained upon expression of cytoplasmic NES-GA$_{65}$-GFP. However, in the presence of aggregates of NLS-GA$_{65}$-GFP, NLS-LS failed to efficiently repartition to the nucleoplasm (*Figure 2C*, *Figure 2—figure supplement 1A*), indicating that the nuclear poly-GA aggregates compromise nucleolar protein quality control.

We next analyzed the viability of HEK293 cells expressing poly-GA in cytoplasm or nucleus. We used the MTT assay to measure metabolic activity, which reflects not only cytotoxic effects but also changes in cell division and overall cellular fitness. Cytoplasmic and nuclear forms of an

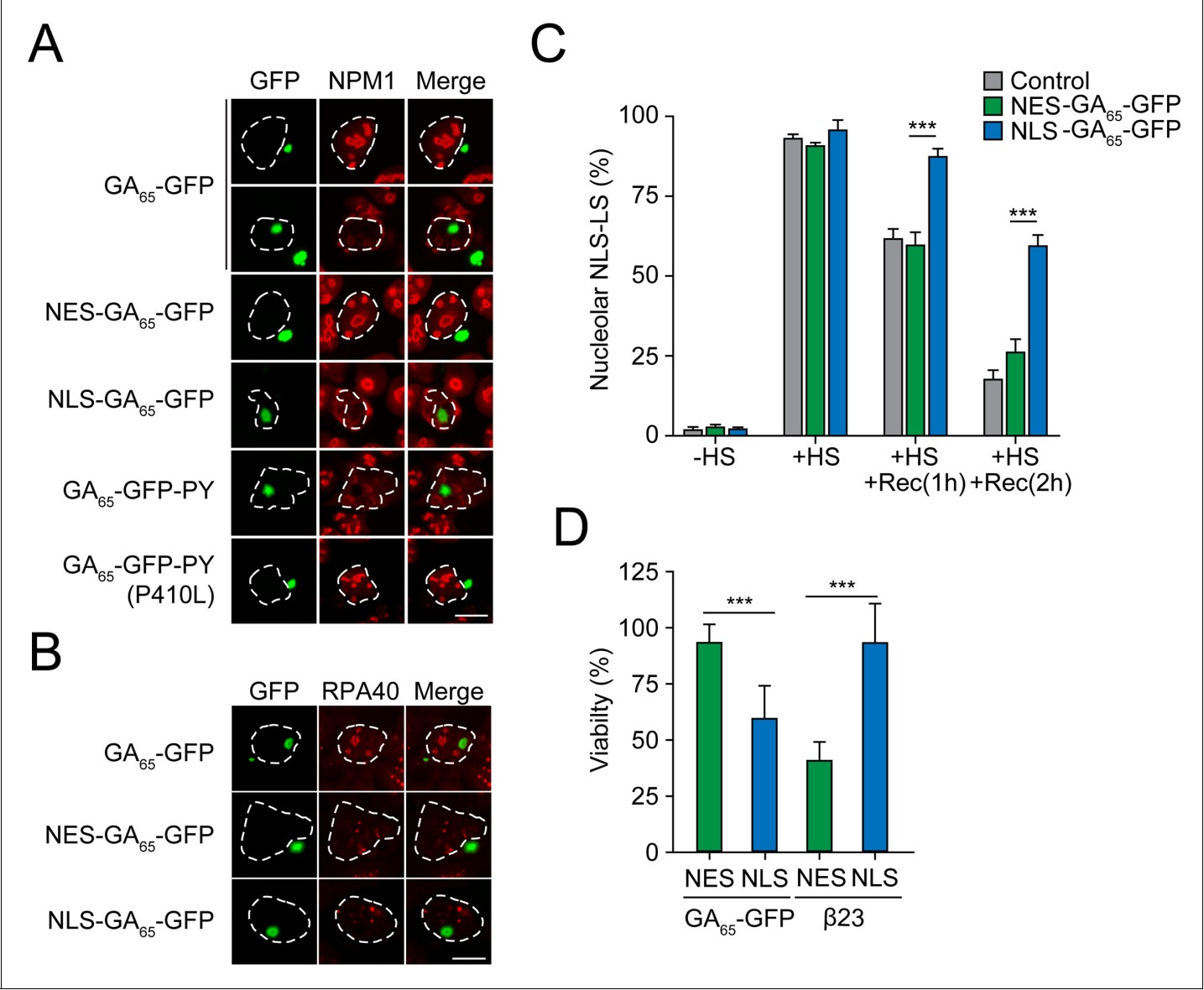

**Figure 2.** Nuclear poly-GA aggregates compromise nucleolar integrity and impair cell viability. (A) Nuclear poly-GA aggregates alter nucleophosmin (NPM1) localization. The indicated constructs were transfected into HEK293 cells. Cells were fixed and stained with anti-NPM1 antibodies (red). Poly-GA was visualized by GFP fluorescence (green). (B) Nuclear poly-GA aggregates are not nucleolar. The indicated constructs were transfected into HEK293 cells, followed by staining with antibodies against DNA-directed RNA polymerases I and III subunit RPAC1 (RPA40) (red). Poly-GA was visualized by GFP fluorescence (green). (C) Nuclear poly-GA aggregates disrupt nucleolar protein quality control. HEK293 cells were co-transfected with NLS-firefly luciferase fused to mScarlet (NLS-LS) and the indicated poly-GA constructs or GFP as a control. Cells were maintained at 37°C (–HS) or subjected to heat stress 43°C (+HS) for 2 hr or heat stress and recovery (+HS + Rec) for 1 hr and 2 hr. Cells with nucleolar NLS-LS were counted, and the results plotted as percentage of transfected cells. Data are shown as mean + SD (n = 3). p-Value of two-sided t-test is displayed (***p≤0.001). Representative immunofluorescence images are shown in *Figure 2—figure supplement 1A*. (D) Nuclear poly-GA is toxic. HEK293 cells were transfected with the indicated constructs, and MTT cell viability assays were performed 4 days after transfection. Data were normalized to cells transfected with empty vector. Data are shown as means + SD (n ≥ 3); p-values of two-sided t-test are shown (***p≤0.001). White dashed lines delineate the nucleus based on DAPI staining. Scale bars represent 10 μm.

The online version of this article includes the following source data and figure supplement(s) for figure 2:

**Source data 1.** Numerical values for graph in *Figure 2C*.
**Source data 2.** Numerical values for graph in *Figure 2D*.
**Figure supplement 1.** Nuclear poly-GA aggregates impair nucleolar protein quality control and cell viability.

amyloidogenic model protein without repeat sequences (β23) served as a control (*West et al., 1999*; *Woerner et al., 2016*). While the expression of NES-GA$_{65}$-GFP did not cause toxicity, metabolic activity was significantly reduced upon expression of NLS-GA$_{65}$-GFP (*Figure 2D*). This effect was reproduced in cells transfected with a plasmid coding for GA$_{65}$ from an alternative degenerated and G$_4$C$_2$-free DNA sequence (GA$_{65}$(2)) (*Figure 1—figure supplement 1A*, *Figure 2—figure supplement 1B*), further excluding RNA-mediated toxicity. Decreased viability was also observed when GA$_{65}$-GFP was targeted to the nucleus via the alternative PY localization signal (*Figure 2—figure*

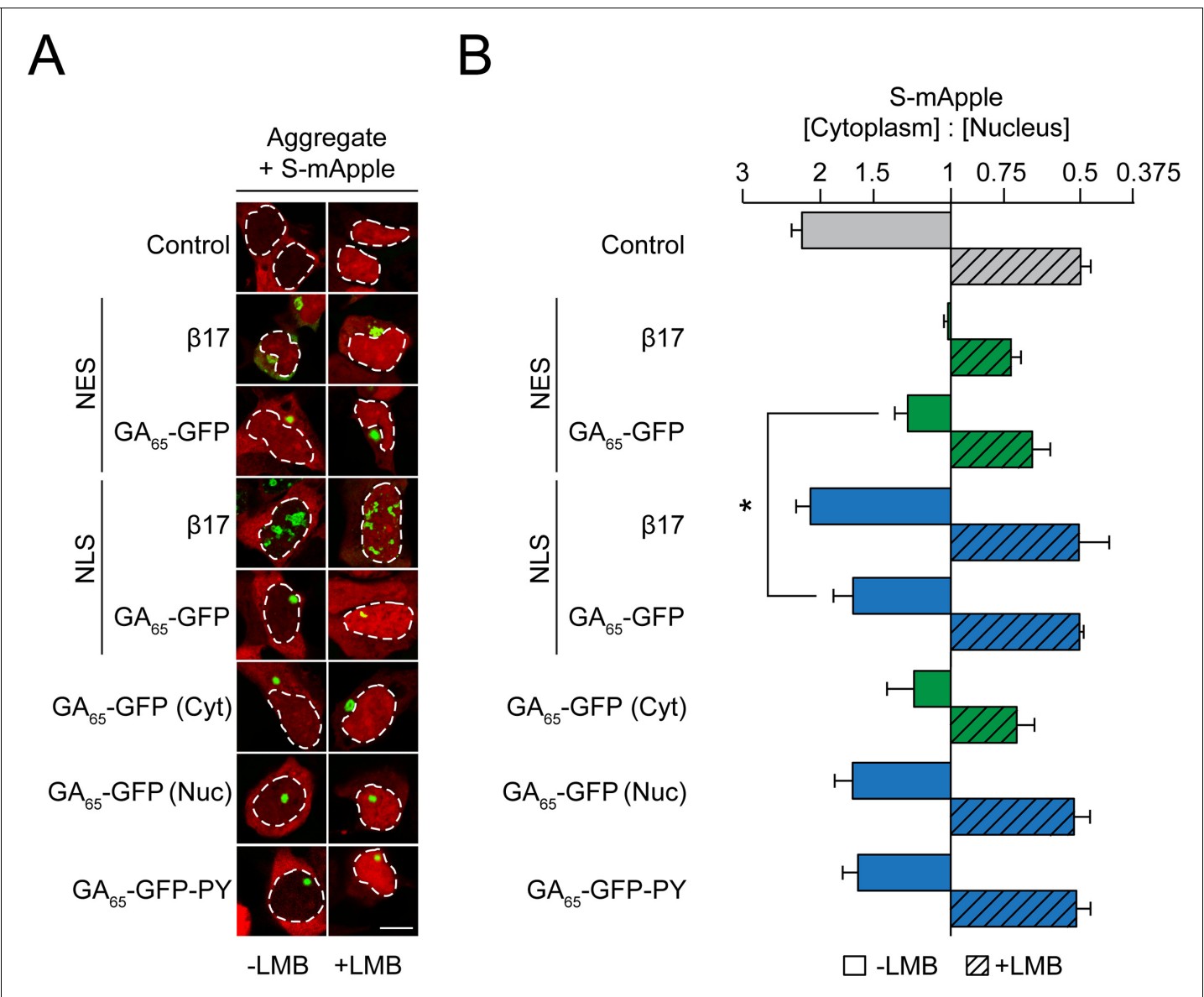

**Figure 3.** Cytoplasmic poly-GA aggregates impair nucleocytoplasmic protein transport. (**A**) Cytoplasmic poly-GA aggregates alter nuclear transport of a shuttling reporter protein. HEK293 cells were co-transfected with S-mApple (red) and either empty vector (Control), NES-β17, NES-GA$_{65}$-GFP, NLS-β 17, NLS-GA$_{65}$-GFP, GA$_{65}$-GFP, or GA$_{65}$-GFP-PY (green). Leptomycin B (LMB; 10 ng/ml) was added for 15 min when indicated. White dashed lines delineate nuclei based on DAPI staining. Scale bar represents 10 μm. (**B**) Quantification of S-mApple distribution from data in (**A**). The x-axis shows the enrichment of S-mApple concentration in the cytoplasm relative to the nucleus. Cells transfected with GA$_{65}$-GFP were further analyzed and divided into cells with cytoplasmic (Cyt) or nuclear (Nuc) aggregates. Data are means + SD, n = 3 independent experiments, >70 cells were analyzed per condition. *p≤0.05 from two-sided t-test.

The online version of this article includes the following source data for figure 3:

**Source data 1.** Numerical values for graph in *Figure 3B*.

supplement 1C). Importantly, both NES-GA$_{65}$-GFP and NLS-GA$_{65}$-GFP were expressed at levels comparable to GA$_{65}$-GFP (*Figure 2—figure supplement 1D*). Point mutations in the NLS or PY targeting sequence prevented accumulation of GA$_{65}$-GFP in the nucleus and restored viability (*Figure 2—figure supplement 1C*). Together, these results indicate that the difference in toxicity between nuclear and cytoplasmic poly-GA is caused by compartment-specific properties of the aggregates independent of their targeting sequences and mRNA.

## Cytoplasmic GA$_{65}$-GFP aggregates interfere with nuclear transport

We have previously shown that artificial β-sheet proteins, when aggregating in the cytoplasm, sequester nuclear transport factors and thereby interfere with transport of proteins and mRNA across the nuclear envelope (*Woerner et al., 2016*). Similar observations were made for the aggregates of various disease proteins including poly-GA and R-DPRs (*Boeynaems et al., 2016*; *Chou et al., 2018*; *Eftekharzadeh et al., 2019*; *Freibaum et al., 2015*; *Gasset-Rosa et al., 2017*; *Grima et al., 2017*; *Jovičić et al., 2015*; *Khosravi et al., 2017*; *Kramer et al., 2018*; *Solomon et al., 2018*; *Zhang et al., 2015*; *Zhang et al., 2016*). To test which role the localization of the poly-GA proteins plays in this process, we expressed NES-GA$_{65}$-GFP or NLS-GA$_{65}$-GFP together with the reporter protein shuttle-mApple (S-mApple). This reporter protein contains both nuclear import and export signals and consequently shuttles between nucleus and cytoplasm. At steady state, S-mApple localized mainly to the cytoplasm, but accumulated within minutes in the nucleus upon inhibition of nuclear export with leptomycin B (LMB) (*Wolff et al., 1997*; *Figure 3*).

As previously described, S-mApple was retained within the nucleus upon expression of the cytoplasmic β-sheet protein NES-β17 (*Figure 3*), indicative of inhibition of nuclear protein export (*Woerner et al., 2016*). Expression of NES-GA$_{65}$-GFP also impaired S-mApple export from the nucleus (-LMB), but to a lesser extent than NES-β17 (*Figure 3*). A mild impairment of nuclear protein import by NES-GA$_{65}$-GFP was also observed, as measured upon inhibition of export with LMB (*Figure 3*). As for NLS-β17, nuclear poly-GA aggregates had only a weak effect on protein export (*Figure 3*). Similarly, cells containing nuclear aggregates did not show a significant change in S-mApple import into the nucleus (*Figure 3*). These findings were replicated in cells displaying cytoplasmic or nuclear poly-GA aggregates of untargeted GA$_{65}$-GFP (*Figure 3*).

We next monitored the nuclear translocation of p65, a subunit of the NF-KB complex, upon stimulation by the cytokine TNFα. In control cells, p65 is largely cytoplasmic and enters the nucleus upon treatment with TNFα (*Figure 4A, B*). Cells containing NES-GA$_{65}$-GFP aggregates displayed a potent inhibition of p65 translocation (*Figure 4A, B*). A similar effect was seen with cytoplasmic aggregates of polyQ-expanded Huntingtin-exon 1 (Htt96Q) as a positive control (*Figure 4A, B*; *Woerner et al., 2016*). Cells containing cytoplasmic aggregates of untargeted GA$_{65}$-GFP also displayed reduced p65 translocation (*Figure 4A, B*). The observed translocation impairment was independent of an alteration of p65 phosphorylation and degradation of the inhibitor of nuclear factor κB (IκB) (*Figure 4C*). In contrast, nuclear poly-GA aggregates showed only a limited effect on p65 translocation (*Figure 4A, B*).

Cytoplasmic aggregates of β-sheet model proteins and disease-linked, amyloidogenic proteins cause mislocalization and sequestration of nuclear pore complexes and importins (*Woerner et al., 2016*). Particularly, R-DPRs have also been shown to directly bind and interfere with cargo loading onto importin β at the nuclear pore (*Hayes et al., 2020*). While cytoplasmic poly-GA aggregates had no apparent effect on the localization of the nuclear pore complex (*Figure 1B*), we found that cells with cytoplasmic GA$_{65}$-GFP aggregates frequently contained aggregate foci of importins α1 (KPNA2) and α3 (KPNA4), as has been observed previously for cytoplasmic aggregates of the aggregation-prone model proteins NES-β23 (*Woerner et al., 2016*). This effect was also observed for importin β1 (KPNB1), although to a lesser degree (*Figure 4—figure supplement 1*). Additionally, importins appeared to be enriched at the periphery of poly-GA inclusions, as seen for KPNA2 and KPNA4 (*Figure 4—figure supplement 1*). Nuclear poly-GA aggregates had no effect on the distribution of these importins (*Figure 4—figure supplement 1*). Thus, similar to the artificial β-sheet proteins, poly-GA aggregates induce compartment-specific cellular defects and impair nucleocytoplasmic protein transport.

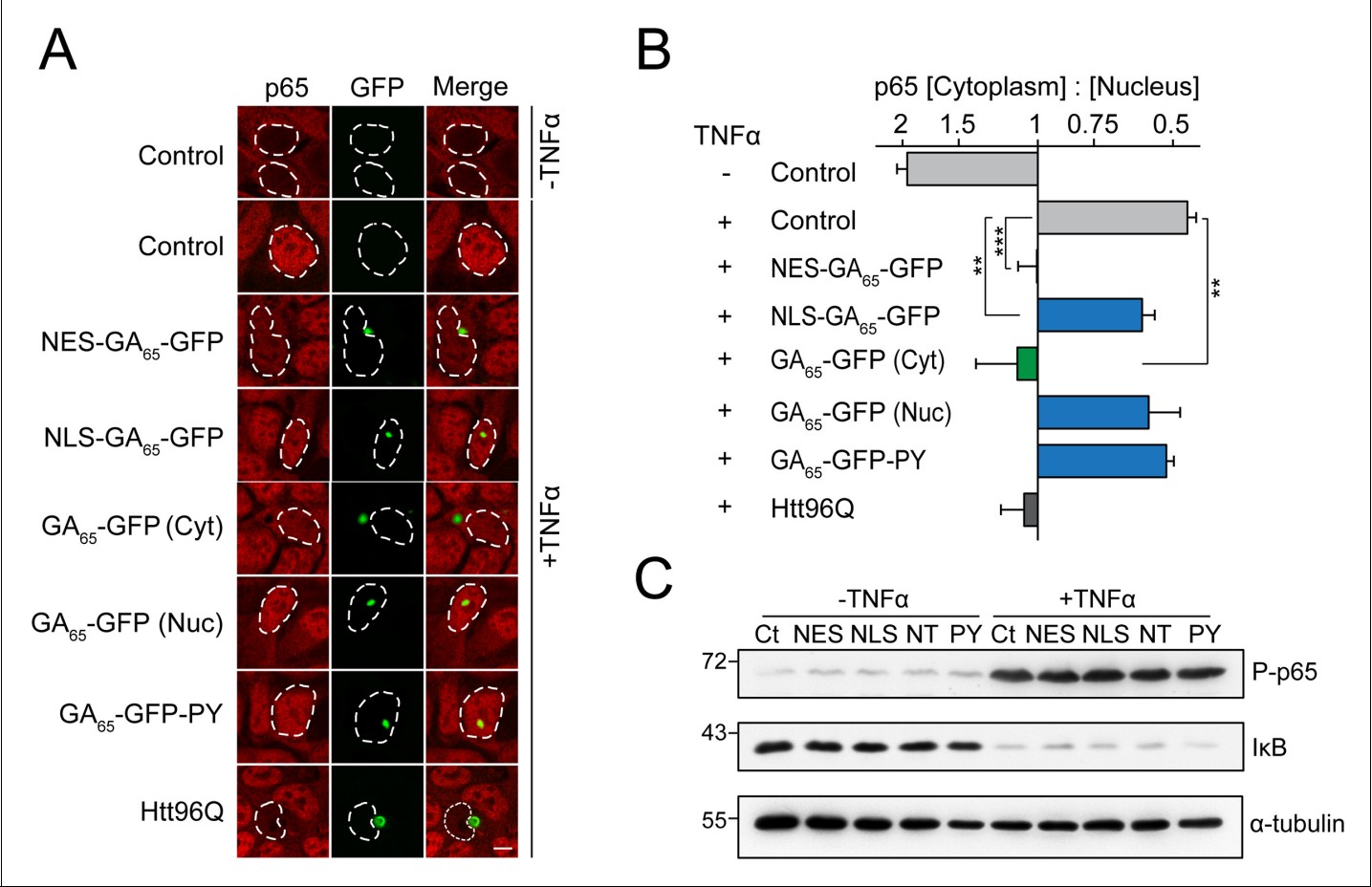

**Figure 4.** Cytoplasmic poly-GA aggregates inhibit nuclear import of p65. (**A**) Cytoplasmic poly-GA aggregates inhibit p65 nuclear translocation. HEK293 cells were transfected with empty vector (Control), NES-GA$_{65}$-GFP, NLS-GA$_{65}$-GFP, GA$_{65}$-GFP, GA65-GFP-PY, or Htt96Q-GFP (Htt96Q) (green) and analyzed for NF-κB p65 localization (red) with and without TNFα treatment (30 min). White dashed lines delineate nuclei based on DAPI staining. Scale bar represents 10 μm. (**B**) Quantification of NF-κB p65 distribution from data in (**A**). The x-axis shows the enrichment of p65 in the cytoplasm relative to the nucleus. Data are means + SD (n = 3), >100 cells were analyzed per condition. **p≤0.01, ***p≤0.001 from two-sided t-test. (**C**) Expression of poly-GA does not alter the degradation of IκB and phosphorylation of p65. HEK293 cells were transfected with the indicated constructs (Ct: Control; NES: NES-GA$_{65}$-GFP; NLS: NLS-GA$_{65}$-GFP; NT: GA$_{65}$-GFP; PY: GA$_{65}$-GFP-PY) and treated as described in (**A**). Levels of IκB and phosphorylated NF-κB p65 (P–p65) were analyzed by immunoblotting. α-tubulin served as loading control.

The online version of this article includes the following source data and figure supplement(s) for figure 4:

**Source data 1.** Numerical values for graph in *Figure 4B*.

**Figure supplement 1.** Importins form aggregates in cells containing cytoplasmic poly-GA aggregates and are partially sequestered.

## G$_4$C$_2$ repeat mRNA causes nuclear mRNA retention and pronounced toxicity

An aberrant distribution of mRNA has been observed in mouse motor neuron-like cells expressing expanded G$_4$C$_2$ repeats and in *C9orf72* patient cortical neurons (*Freibaum et al., 2015*; *Rossi et al., 2015*). We used an oligo-dT probe to test whether cytoplasmic poly-GA aggregates, generated from constructs lacking G$_4$C$_2$, affect the cellular distribution of total mRNA. In control cells, mRNA was present throughout the cytoplasm and in small nuclear ribonucleic particles (*Figure 5A*; *Carter et al., 1991*). Expression of NES-GA$_{65}$-GFP had only a minor effect on cellular mRNA distribution (*Figure 5A, B*), in contrast to the expression of cytoplasmic β-protein (NES-β17), which resulted in pronounced nuclear mRNA retention (*Woerner et al., 2016*). Likewise, NLS-GA$_{65}$-GFP only caused limited mRNA retention in the nucleus (*Figure 5A, B*). Thus, poly-GA aggregates interfere with the function of only a subset of nuclear transport factors.

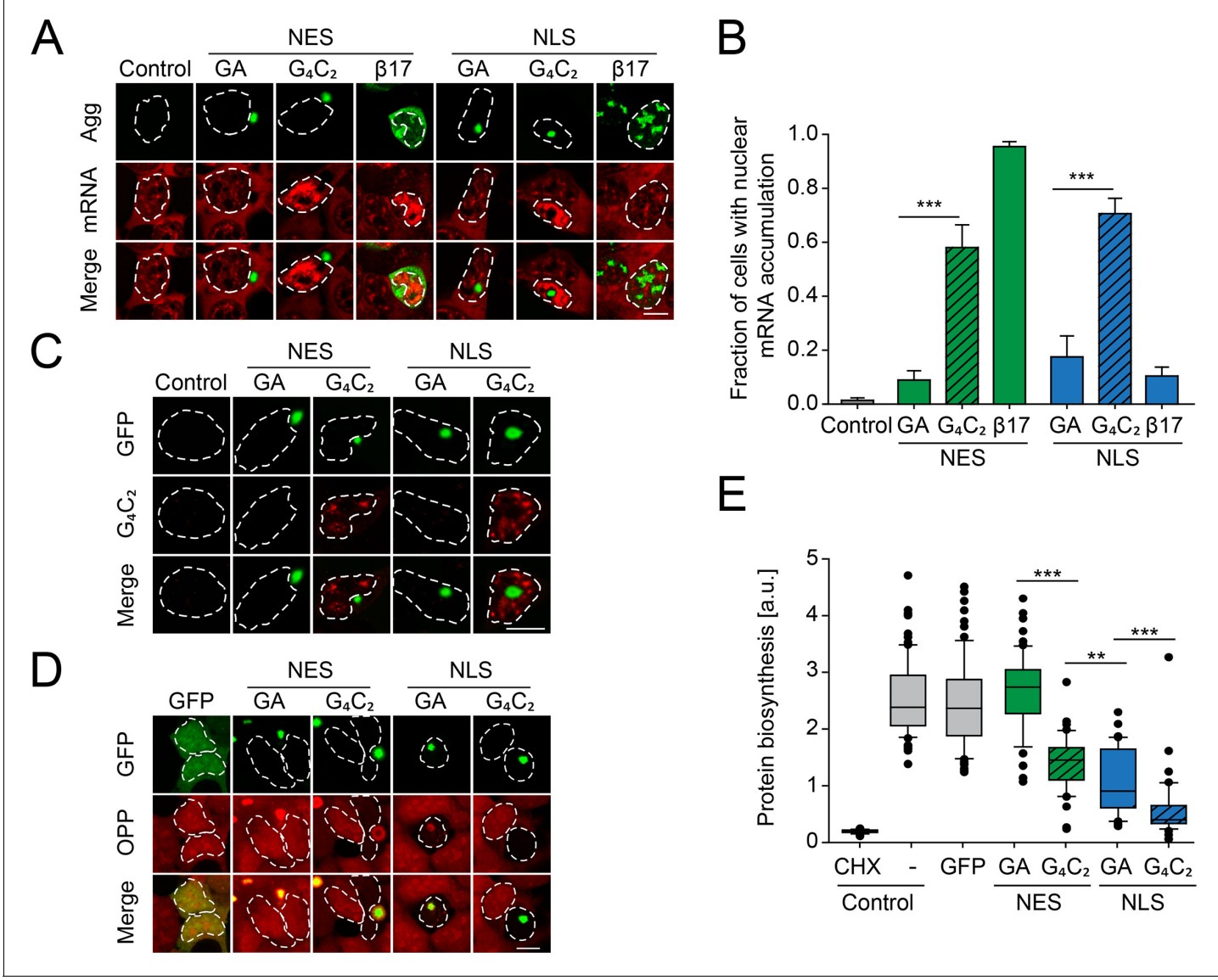

**Figure 5.** Protein biosynthesis defects correlate with retention of mRNA in the nucleus induced by $G_4C_2$ mRNA as well as presence of nuclear poly-GA aggregates. (**A**) $G_4C_2$-containing constructs induce strong nuclear mRNA accumulation. HEK293 cells were transfected with the indicated constructs: empty vector (Control); NES-$GA_{65}$-GFP (NES-GA); NES-$G_4C_2$-GFP (NES-$G_4C_2$); NES-β17; NLS-$GA_{65}$-GFP (NLS-GA); NLS-$G_4C_2$-GFP (NLS-$G_4C_2$); NLS-β17. PolyA-RNA was detected by fluorescence in situ hybridization using a poly-dT probe (red); protein aggregates (Agg) green. (**B**) Quantification of data in (**A**). The graph shows the fraction of cells with nuclear mRNA accumulation. Data are means + SD (n = 3), ***p≤0.001 from two-sided t-test. (**C**) $G_4C_2$-containing constructs induce the formation of $G_4C_2$ RNA foci. HEK293 cells were transfected with the indicated constructs: empty vector (Control); NES-$GA_{65}$-GFP (NES-GA); NES-$G_4C_2$-GFP (NES-$G_4C_2$); NLS-$GA_{65}$-GFP (NLS-GA); NLS-$G_4C_2$-GFP (NLS-$G_4C_2$). Cells were analyzed for GFP fluorescence (green) and $C_4G_2$ fluorescence by in situ hybridization (red). (**D**) Decreased protein biosynthesis in the presence of nuclear poly-GA and $G_4C_2$ mRNA. Newly synthesized proteins were labeled with O-propargyl-puromycin (OPP; red) in HEK293 cells transfected with the indicated constructs (green). The white dashed lines delineate the nucleus based on DAPI staining, and the scale bar represents 10 μm. (**E**) Quantification of data in (**D**). Analysis of control cells transfected with empty vector and treated with the translation inhibitor cycloheximide (CHX) when indicated is included as control. Boxplot of a representative experiment is shown. Center lines show the medians; box limits indicate the 25th and 75th percentiles; whiskers extend to the 10th and 90th percentiles, outliers are plotted as circles. Welch's t-test was used to assess statistical significance (**p≤0.01; ***p≤0.001).

The online version of this article includes the following source data and figure supplement(s) for figure 5:

**Source data 1.** Numerical values for graph in *Figure 5B*.

**Source data 2.** Numerical values for repeats for *Figure 5E*.

**Figure supplement 1.** Expression of $G_4C_2$-containing constructs induces mRNA retention in the nucleus.

Given that expression of poly-GA protein alone did not recapitulate the alterations of mRNA localization observed in *C9orf72* patient brain (*Freibaum et al., 2015*; *Rossi et al., 2015*), we next analyzed the effect of an ATG-driven poly-GA construct of 73 GA repeats encoded entirely by $G_4C_2$ motifs $(G_4C_2)_{73}$ (*Figure 1A, Figure 1—figure supplement 1A*). Comparable to the synthetic $GA_{65}$-GFP sequences, these constructs also generated poly-GA protein aggregates in the cytoplasm or nucleus (*Figure 5A, C*). Importantly, as cytoplasmic poly-GA did not induce toxicity in our cell system, the use of NES and NLS targeting sequences allowed us to isolate the contribution of the $G_4C_2$-containing RNA to cellular pathology. Both NES-$(G_4C_2)_{73}$-GFP and NLS-$(G_4C_2)_{73}$-GFP, besides generating DPR inclusions, resulted in the formation of $G_4C_2$-positive RNA foci in the nucleus, as observed by fluorescent in situ hybridization (FISH) using a $C_4G_2$ probe (*Figure 5C*). Cells expressing $GA_{65}$-GFP or $(G_4C_2)_{73}$-GFP without targeting sequence were analyzed as well, but only the $G_4C_2$ constructs showed mRNA accumulation in the nucleus of the majority of cells (*Figure 5—figure supplement 1A*). Furthermore, simultaneous visualization of $G_4C_2$ RNA puncta with the $C_4G_2$ probe, total mRNA using an oligo-dT probe and GA-GFP revealed that the $G_4C_2$ RNA foci are associated with nuclear mRNA accumulation, independent of the presence of visible poly-GA protein aggregates (*Figure 5—figure supplement 1B*). Together, these results indicate that $G_4C_2$ RNA, not poly-GA protein, mediates retention of mRNA in the nucleus.

The nuclear mRNA retention in cells expressing $G_4C_2$ constructs was accompanied by a marked reduction of protein synthesis as measured by the incorporation of a puromycin derivative into newly translated proteins (*Slomnicki et al., 2016*; *Figure 5D, E*). Interestingly, NLS-$GA_{65}$-GFP-expressing cells also displayed reduced protein synthesis, independently of $G_4C_2$ RNA. In contrast, NES-$GA_{65}$-GFP had no inhibitory effect on protein synthesis (*Figure 5D, E*). Notably, cells expressing NLS-$(G_4C_2)_{73}$-GFP, accumulating both nuclear poly-GA protein and $G_4C_2$ RNA, were almost completely translation inactive, similar to control cells treated with the translation inhibitor cycloheximide (*Figure 5D, E*). Thus, both nuclear poly-GA protein and $G_4C_2$ RNA appear to have additive inhibitory effects on protein biosynthesis. We next measured the proliferation rate or viability of cells transiently transfected with either synthetic or $G_4C_2$-containing constructs. Independent of the presence of a NES or NLS targeting sequence, all $G_4C_2$ constructs markedly decreased cellular viability (*Figure 6A*), indicating that toxicity was mediated by the expanded $G_4C_2$ RNA. Moreover, expression of NLS-$(G_4C_2)_{73}$-GFP was more toxic than NLS-$GA_{65}$-GFP (*Figure 6A*), although the poly-GA levels of the constructs were comparable (*Figure 6—figure supplement 1A*).

Long sequences of repeated $G_4C_2$ motifs can produce a series of different DPRs by RAN translation (*Ash et al., 2013*; *Gendron et al., 2013*; *Mori et al., 2013a*; *Mori et al., 2013c*; *Zu et al., 2013*). Since the R-DPRs (poly-PR and poly-GR) have been reported to be toxic in cellular models (*Boeynaems et al., 2016*; *Freibaum et al., 2015*; *Lee et al., 2016*; *Shi et al., 2017*; *Tao et al., 2015*; *Zu et al., 2013*), we investigated whether the $G_4C_2$ constructs produced R-DPRs at levels sufficient to explain the observed toxicity independently of $G_4C_2$ RNA effects. To this end, we engineered synthetic sequences, resulting in the ATG-driven synthesis of 73 GR or 73 PR repeats ($GR_{73}$-GFP and $PR_{73}$-GFP, respectively) without $G_4C_2$ repeats. $GR_{73}$-GFP was located predominantly within the cytoplasm and the nucleolus of HEK293 cells, while $PR_{73}$-GFP accumulated within the nucleoplasm and nucleolus (*Figure 6—figure supplement 1B*), as reported previously (*Frottin et al., 2019*; *Lee et al., 2016*; *May et al., 2014*; *White et al., 2019*). While the expression of $(G_4C_2)_{73}$-GFP dramatically reduced cellular viability, the R-DPRs were not measurably cytotoxic (*Figure 6A*), consistent with an earlier report for constructs with longer repeat lengths (*May et al., 2014*). We used specific anti-GR and anti-PR antibodies to determine the relative accumulation of the R-DPRs in $(G_4C_2)_{73}$-GFP-expressing cells in comparison to cells expressing $GR_{73}$-GFP and $PR_{73}$-GFP. The antibodies recognized specific signals in cells transfected with the respective R-DPR constructs, but failed to detect R-DPR protein in cells expressing $(G_4C_2)_{73}$-GFP (*Figure 6B*), indicating that production of R-DPR protein from the $G_4C_2$ constructs was very inefficient. Given that $PR_{73}$-GFP and $GR_{73}$-GFP were produced in detectable quantities from the synthetic constructs without inducing toxicity, we conclude that the pronounced toxicity observed upon expression of $(G_4C_2)_{73}$-GFP cannot be explained by RAN translation of R-DPR but is due to nuclear mRNA retention mediated by the $G_4C_2$ repeat RNA.

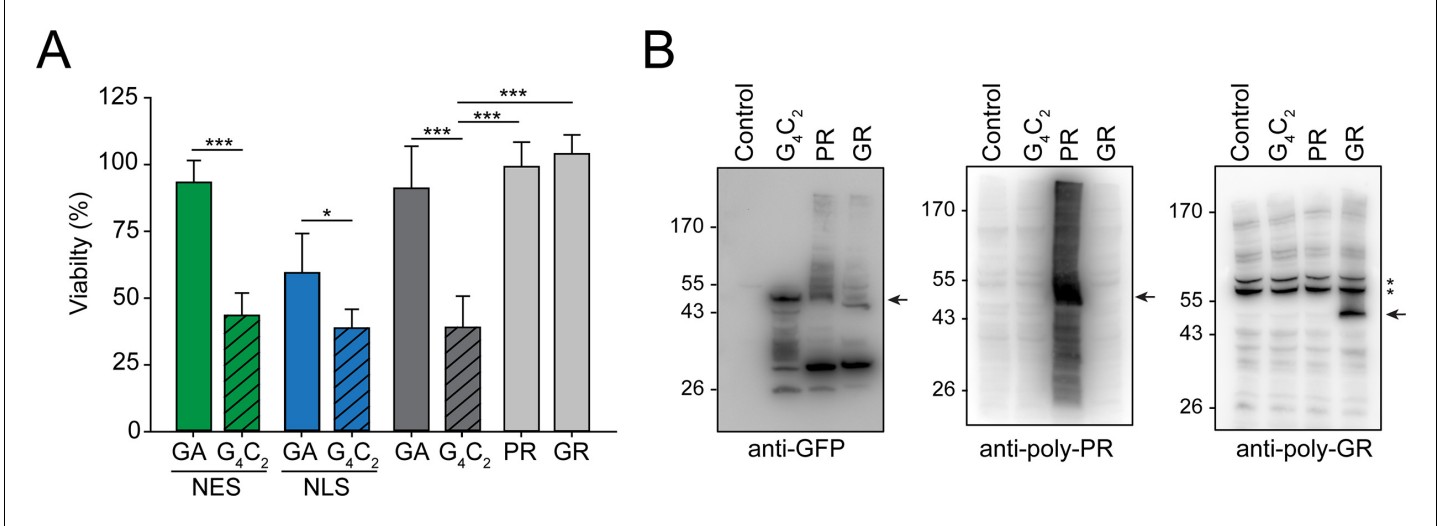

**Figure 6.** Production of $G_4C_2$ mRNA strongly decreases cellular viability. (**A**) $G_4C_2$ mRNA induces strong toxicity. HEK293 cells were transfected with the indicated constructs: $G_4C_2$-GFP ($G_4C_2$), NES-$G_4C_2$-GFP (NES-$G_4C_2$), NLS-$G_4C_2$-GFP (NLS-$G_4C_2$), $GA_{65}$-GFP (GA), NES-$GA_{65}$-GFP (NES-GA), NLS-$GA_{65}$-GFP (NLS-GA), $PR_{73}$-GFP (PR), or $GR_{73}$-GFP (GR). MTT cell viability assays were performed 4 days after transfection. Data were normalized to cells transfected with empty vector. Data are means + SD (n ≥ 3). Part of this data is also shown in *Figure 2D*. *p≤0.05; ***p≤0.001 from two-sided t-test. (**B**) $(G_4C_2)_{73}$-GFP does not produce detectable amounts of arginine containing dipeptide repeats (R-DPRs). HEK293 cells were transfected with the indicated constructs: empty vector (control), $(G_4C_2)_{73}$-GFP ($G_4C_2$), $PR_{73}$-GFP (PR), and $GR_{73}$-GFP (GR). Immunoblot analysis was then performed against GFP (left), poly-PR (center), and poly-GR (right). A representative result of three biological repeats is shown. The arrows indicate the main band of the respective DPRs, and * indicate non-specific bands recognized by the anti-GR antibody.

The online version of this article includes the following source data and figure supplement(s) for figure 6:

**Source data 1.** Numerical values for graph in *Figure 6A*.
**Figure supplement 1.** Expression levels of GA-DPRs and localization of R-DPRs.

## Discussion

We have employed a cellular model to differentiate possible mechanisms of toxicity exerted by expansion of the $G_4C_2$ hexanucleotide tract within the *C9orf72* locus, the most frequent genetic cause of ALS and FTD (*DeJesus-Hernandez et al., 2011*; *Renton et al., 2011*). Our results demonstrated that the $G_4C_2$ expansion causes toxicity in our cellular system in a manner dependent on both aggregates of $G_4C_2$-encoded DPR proteins and the $G_4C_2$ repeat mRNA. These findings suggest a multiple hit model of additive, but mechanistically independent, effects of proteotoxicity and RNA-mediated toxicity, with the latter being the predominant toxic agent in the model system investigated. However, the longer $G_4C_2$ expansions, and the resulting DPRs, present in patients may have additional adverse effects that were not recapitulated with the constructs used here.

Both DPRs and repeat mRNA interfered with different aspects of nucleocytoplasmic transport. The $G_4C_2$ RNA inhibited the export of mRNA from the nucleus, consistent with a dramatic impairment of protein synthesis that would be associated with strong neuronal toxicity. Notably, this effect was independent of the presence of DPR protein aggregates, which is in agreement with results from zebrafish (*Swinnen et al., 2018*), but in contrast to findings observed in *Drosophila* (*Tran et al., 2015*). Additionally, as has been reported for artificial β-sheet proteins and disease-associated aggregation-prone proteins such as Tau, Huntingtin, FUS, and TDP-43 (*Chou et al., 2018*; *Dormann et al., 2010*; *Eftekharzadeh et al., 2019*; *Gasset-Rosa et al., 2017*; *Grima et al., 2017*; *Woerner et al., 2016*), cytoplasmic poly-GA aggregates partially inhibited the transport of proteins across the nuclear pore. This observation is in agreement with the previously reported sequestration of a subset of transport factors by poly-GA (*Khosravi et al., 2017*; *Solomon et al., 2018*; *Zhang et al., 2016*), and the importin pathology that has been observed in postmortem

frontal cortex tissue of ALS/FTD *C9orf72* patients (*Solomon et al., 2018*). Interestingly, other DPRs have also been reported to inhibit additional aspects of nucleocytoplasmic transport (*Boeynaems et al., 2017*; *Cook et al., 2020*; *Lee et al., 2016*; *Lin et al., 2016*; *Zhang et al., 2018a*). Poly-PR can directly obstruct the central channel of the nuclear pore by binding to the FG domains of nuclear pore proteins (*Shi et al., 2017*), and R-DPRs can disrupt cargo loading onto karyopherins β (*Hayes et al., 2020*).

While the effects of cytoplasmic poly-GA (NES-poly-GA) aggregates on nuclear transport were well tolerated in our cell culture model, poly-GA aggregates within the nucleus (NLS-poly-GA) were associated with substantial proteotoxicity. Nuclear poly-GA formed aggregates at sites that were distinct from nucleoli, similar to the localization of poly-GA aggregates observed in patient brain (*Schludi et al., 2015*). However, formation of these inclusions altered the shape of nucleoli as a result of mislocalization and partial sequestration of the abundant GC protein NPM1, consistent with the nucleolar abnormalities reported in neurons from C9orf72 cases (*Mizielinska et al., 2017*). Nuclear poly-GA aggregates interfered with the recently described protein quality control function of the GC phase of the nucleolus (*Frottin et al., 2019*), as demonstrated using the metastable firefly luciferase as a model substrate. In cells containing nuclear poly-GA aggregates, misfolded luciferase failed to repartition from the nucleolus to the nucleoplasm during recovery from stress. This inhibitory effect was comparable to that previously observed for positively charged DPRs, such as poly-PR, which accumulate directly within the GC phase of the nucleolus (*Frottin et al., 2019*; *Mizielinska et al., 2017*; *Tao et al., 2015*). However, R-DPRs apparently do not accumulate in the nucleolus in patient brain (*Mackenzie et al., 2015*; *Schludi et al., 2015*), and thus may rather interfere with nucleolar quality control by the mechanism described here for nuclear poly-GA (*Frottin et al., 2019*; *Kwon et al., 2014*; *White et al., 2019*).

Expression of poly-GA from $G_4C_2$-containing constructs resulted in substantially greater toxicity than expression from synthetic constructs when similar poly-GA lengths and amounts were compared. The $G_4C_2$ hexanucleotide repeat is thought to exert toxic effects in part by forming higher-order RNA assemblies in the nucleus that sequester RNA-binding proteins (*Almeida et al., 2013*; *Cooper-Knock et al., 2015*; *Cooper-Knock et al., 2014*; *Donnelly et al., 2013*; *Haeusler et al., 2014*; *Mori et al., 2013b*; *Reddy et al., 2013*; *Rossi et al., 2015*; *Sareen et al., 2013*; *Xu et al., 2013*). Even if longer $G_4C_2$ sequences may mediate additional effects (*West et al., 2020*), the $G_4C_2$ constructs tested here resulted in a pronounced inhibition of protein synthesis, even when coding for cytoplasmic poly-GA protein (NES-GA$_{65}$), which did not impair protein biogenesis when produced from a synthetic (non-$G_4C_{2)}$ construct. However, because expression constructs based on $G_4C_2$ repeats also generate the different RAN translation DPR products, a clear distinction between RNA and DPR toxicity in previous studies had been difficult. We therefore compared not only the relative toxicity of poly-GA and $G_4C_2$ constructs, but also measured the toxicity of poly-PR and poly-GR constructs in the absence of $G_4C_2$ repeat RNA. Notably, expression of these protein-only constructs did not induce overt toxicity even when expressed at levels much higher than those generated by RAN translation of the $G_4C_2$ repeat. These findings allowed us to unequivocally attribute the major component of toxicity associated with $G_4C_2$ constructs to the production of the pathological $G_4C_2$ RNA sequence. Although DPRs may be undetectable in the brain regions most affected by neurodegeneration in *C9orf72* patients (*Schludi et al., 2015*), we cannot exclude a DPR contribution to pathology in carriers of $G_4C_2$ expansions, where combinations of the various DPRs and mRNA are present simultaneously, and nondividing cells are exposed to these factors for extended time periods. Indeed, nuclear poly-GA also interfered with protein biosynthesis, presumably by impairing the nucleolar function in ribosome biogenesis, and thus could enhance the toxic effects of $G_4C_2$ repeat RNA.

In summary, our results indicate that the $G_4C_2$ expansion in *C9orf72* interferes with multiple nuclear functions, culminating in an inhibition of protein biogenesis, an outcome that would be especially harmful to neuronal cells. These dominant toxic effects might be further aggravated by a loss of function of the endogenous C9ORF72 protein, which is thought to play a role in cellular quality control (*Boivin et al., 2020*; *Sellier et al., 2016*; *Sullivan et al., 2016*; *Yang et al., 2016*; *Zhu et al., 2020*). Further research on the relative contribution of the different toxic mechanisms will be important in developing therapeutic strategies.

# Materials and methods

## Key resources table

| Reagent type (species) or resource | Designation | Source or reference | Identifiers | Additional information |
|---|---|---|---|---|
| Cell line (*Homo sapiens*, female) | HEK293 | ATCC | Cat. #: ATCC-CRL-1573 RRID:CVCL_0045 | Lot/Batch No: 63777489 |
| Recombinant DNA reagent | pcDNA3.1 GA$_{65}$-GFP | This paper | | Described in Results part 1 and Materials and methods section |
| Recombinant DNA reagent | pcDNA3.1 NES-GA$_{65}$-GFP | This paper | | Described in Results part 1 and Materials and methods section |
| Recombinant DNA reagent | pcDNA3.1 NLS-GA$_{65}$-GFP | This paper | | Described in Results part 1 and Materials and methods section |
| Recombinant DNA reagent | pcDNA3.1 NLS-GA$_{65}$-GFP (K7T, K15T) | This paper | | Described in Results part 1 and Materials and methods section |
| Recombinant DNA reagent | pcDNA3.1 GA$_{65}$-GFP-PY | This paper | | Described in Results part 1 and Materials and methods section |
| Recombinant DNA reagent | pcDNA3.1 GA$_{65}$-GFP-PY (P410L) | This paper | | Described in Results part 1 and Materials and methods section |
| Recombinant DNA reagent | pcDNA3.1 GFP | This paper | | Described in Results part 1 and Materials and methods section |
| Recombinant DNA reagent | pcDNA3.1 PR$_{73}$-GFP | This paper | | Described in Results part 3 and Materials and methods section |
| Recombinant DNA reagent | pcDNA3.1 GR$_{73}$-GFP | This paper | | Described in Results part 3 and Materials and methods section |
| Recombinant DNA reagent | pcDNA3.1 (G4C2)$_{73}$-GFP | This paper | | Described in Results part 3 and Materials and methods section |
| Recombinant DNA reagent | pcDNA3.1 NLS-(G4C2)$_{73}$-GFP | This paper | | Described in Results part 3 and Materials and methods section |
| Recombinant DNA reagent | pcDNA3.1 NES-(G4C2)$_{73}$-GFP | This paper | | Described in Results part 3 and Materials and methods section |
| Recombinant DNA reagent | Plasmid expressing NLS-LS | PMID:31296649 | | Prof. F. Ulrich Hartl (Max Planck Institute for Biochemistry) |
| Recombinant DNA reagent | Plasmid expressing S-mApple | This paper | | Described in Results part 2 and Materials and methods section |
| Recombinant DNA reagent | Plasmid expressing c-myc-NES-β17 | PMID:26634439 | | Prof. F. Ulrich Hartl (Max Planck Institute for Biochemistry) |
| Recombinant DNA reagent | Plasmid expressing c-myc-NLS-β17 | PMID:26634439 | | Prof. F. Ulrich Hartl (Max Planck Institute for Biochemistry) |
| Recombinant DNA reagent | Plasmid expressing Htt96Q | PMID:26634439 | | Prof. F. Ulrich Hartl (Max Planck Institute for Biochemistry) |

*Continued on next page*

Continued

| Reagent type (species) or resource | Designation | Source or reference | Identifiers | Additional information |
|---|---|---|---|---|
| Antibody | Mouse monoclonal anti-nuclear pore complex proteins | Abcam | Cat. #: ab24609 RRID:AB_448181 | IF (1:1000) |
| Antibody | Mouse monoclonal anti-NPM1 | Invitrogen | Cat. #: 32-5200 RRID:AB_2533084 | IF (1:1000) |
| Antibody | Rabbit monoclonal anti-NF-κB p65 (D14E12) | Cell Signaling Technology | Cat. #: 8242 RRID:AB_10859369 | IF (1:1000) |
| Antibody | Mouse monoclonal anti-p62 | Abcam | Cat. #: ab203430 RRID:AB_2728795 | IF (1:1000) |
| Antibody | Mouse monoclonal anti-c-Myc-Cy3 (9E10) | Sigma | Cat. #: C6594 RRID:AB_258958 | IF (1:1000) |
| Antibody | Rabbit polyclonal anti-amyloid fibrils OC | Millipore | Cat. #: AB2286 RRID:AB_1977024 | IF (1:500) |
| Antibody | Mouse biclonal anti-GFP | Roche | Cat. #: 11814460001 RRID:AB_390913 | WB (1:1000) |
| Antibody | Mouse monoclonal anti-α-tubulin | Sigma | Cat. #: T6199 RRID:AB_477583 | WB (1:1000) |
| Antibody | Mouse monoclonal anti-RPA40 | Santa Cruz | Cat. #: sc-374443 RRID:AB_10991310 | IF (1:1000) |
| Antibody | Mouse monoclonal anti-IκBα (L35A5) | Cell Signaling Technology | Cat. #: 4814 RRID:AB_390781 | WB (1:/1000) |
| Antibody | Rabbit monoclonal anti-phospho-NF-κB p65 (Ser533) (93H1) | Cell Signaling Technology | Cat. #: 3033 RRID:AB_331284 | WB (1:1000) |
| Antibody | Rabbit polyclonal anti-KPNA2 | Abcam | Cat. #: ab70160 RRID:AB_2133673 | IF (1:1000) |
| Antibody | Rabbit polyclonal anti-KPNA4 | Abcam | Cat. #: ab84735 RRID:AB_1860702 | IF (1:1000) |
| Antibody | Mouse monoclonal anti-KPNB1 | Abcam | Cat. #: ab2811 RRID:AB_2133989 | IF (1:1000) |
| Antibody | Mouse monoclonal anti-GAPDH | Millipore | Cat. #: MAB374 RRID:AB_2107445 | WB (1:2000) |
| Antibody | Rabbit polyclonal anti-poly-PR | Proteintech | Cat. #: 23979-1-AP RRID:AB_2879388 | WB (1:1000) |
| Antibody | Mouse monoclonal anti-poly-GR (5A2) | Millipore | Cat. #: MABN778 RRID:AB_2728664 | WB (1:1000) |
| Antibody | Goat Secondary anti-mouse IgG-Alexa 488 | Cell Signaling Technology | Cat. #: 4408 RRID:AB_10694704 | IF (1:1000) |

Continued

| Reagent type (species) or resource | Designation | Source or reference | Identifiers | Additional information |
|---|---|---|---|---|
| Antibody | Goat Secondary anti-mouse IgG-Alexa 555 | Cell Signaling Technology | Cat. #: 4409 RRID:AB_1904022 | IF (1:1000) |
| Antibody | Goat Secondary anti-rabbit IgG-Alexa 555 | Cell Signaling Technology | Cat. #: 4413 RRID:AB_10694110 | IF (1:1000) |
| Antibody | Secondary chicken anti-mouse IgG-Alexa 488 | Invitrogen | Cat. #: 21200 RRID:AB_2535786 | IF (1:1000) |
| Antibody | Secondary goat anti-mouse IgG-Alexa 633 | Invitrogen | Cat. #: A21053 RRID:AB_2535720 | IF (1:1000) |
| Antibody | Secondary goat anti-mouse IgG-peroxidase | Sigma-Aldrich | Cat. #: A4416 RRID:AB_258167 | WB (1:10,000) |
| Antibody | Secondary goat anti-rabbit IgG-peroxidase | Sigma-Aldrich | Cat. #: A9169 RRID:AB_258434 | WB (1:10,000) |
| Sequence-based reagent | $T_{30}$ | PMID:26634439 | FISH probe | Cy5-conjugated |
| Sequence-based reagent | $(C_4G_2)_5$ | This paper | FISH probe | Cy3-conjugated |
| Chemical compound, drug | UltraPure SSC buffer | Thermo Fisher Scientific | Cat. #: 15557044 | For FISH |
| Chemical compound, drug | Formamide | Sigma-Aldrich | Cat. #: 47671 | For FISH |
| Chemical compound, drug | Dextran sulphate | Sigma-Aldrich | Cat. #: D6001 | For FISH |
| Chemical compound, drug | AmyTracker 680 (AmyT) | Ebba Biotech AB | Cat. #: AmyTracker 680 | Amyloid dye |
| Chemical compound, drug | DAPI | Invitrogen | Cat. #: D1306 RRID:AB_2629482 | Nuclear stain |
| Chemical compound, drug | Thiazolyl blue tetrazolium bromide | Sigma Aldrich | Cat. #: M2128 | For cell viability |
| Chemical compound, drug | N,N-dimethylf ormamide | Sigma-Aldrich | Cat. #: D4551 | For cell viability |
| Chemical compound, drug | SDS | Sigma-Aldrich | Cat. #: L3771 | For cell viability |
| Chemical compound, drug | Acetic acid | Sigma-Aldrich | Cat. #: A6283 | For cell viability |
| Chemical compound, drug | Leptomycin B | Sigma Aldrich | Cat. #: L2913 | See Materials and methods section |
| Commercial assay or kit | CellTiter-Glo 2.0 | Promega | Cat. #: G9241 | For cell viability |

*Continued*

| Reagent type (species) or resource | Designation | Source or reference | Identifiers | Additional information |
|---|---|---|---|---|
| Commercial assay or kit | Lipofectamine 2000 transfection reagent | Thermo Fischer Scientific | Cat. #: 11668019 | |
| Commercial assay or kit | Click-iT Plus OPP Alexa Fluor 594 Protein Synthesis Assay Kit | Thermo Fischer Scientific | Cat. #: C10457 | |
| Peptide, recombinant protein | Human recombinant TNFα | Jena Biosciences | Cat. #: PR-430 | |

## Cell culture, transfection, and cell treatments

Human embryonic kidney cells (ATCC-CRL-1573) were obtained from ATCC and maintained in Dulbecco's modified Eagle's medium (DMEM) (Biochrom KG) supplemented with 10% fetal bovine serum (Gibco), 100 U/ml penicillin and 100 µg/ml streptomycin sulfate (Gibco), 2 mM L-glutamine (Gibco), and 5 µg/ml Plasmocin (InvivoGen). Cells were authenticated by DNA fingerprint STR analysis by the supplier and were visually inspected using DAPI DNA staining and tested negative for mycoplasma contamination. For heat stress (HS) and recovery experiments, cells were either maintained at 37°C (-HS), or placed in a 43°C (+HS) incubator for the indicated duration, or subjected to heat stress and then transferred back to 37°C for recovery (+HS + Rec). Transient transfections were performed by electroporation with the GenePulser XCell System (Bio-Rad) or with Lipofectamine 2000 (Invitrogen) according to the manufacturer's instructions. For assessing nuclear import of p65, transfected cells were treated with 20 ng/ml recombinant human TNFα (Jena Biosciences) for 30 min. For translation inhibition, cycloheximide (CHX, Sigma-Aldrich) was dissolved in phosphate buffered saline (PBS) and applied at a final concentration of 1 mM.

## Plasmids

Degenerated sequences encoding 65 GA$^{ggxgcx}$ repeats preceded by a start codon (ATG) and flanked by NheI and BamHI restriction sites were chemically synthesized by GeneArt Gene Synthesis (Invitrogen). The GC content of sequence 1 is 77.2%, while sequence 2 contains 81.5% GC (*Figure 1—figure supplement 1A*). Both GA$_{65}$ sequences were cloned into pcDNA3.1-myc/His A plasmids in frame with GFP at the C-terminus. An N-terminal nuclear export signal (NES) or an N-terminal double SV40 nuclear localization signal (NLS) was inserted by site-directed mutagenesis. The alternative FUS-derived nuclear localization signal (PY) was inserted C-terminally. Point mutations were introduced by site-directed mutagenesis. Similarly, degenerated sequences encoding PR$_{73}$ and GR$_{73}$ were generated by GeneArt Gene Synthesis (Invitrogen) and fused to the N-terminus of GFP. All sequences contained an ATG start codon. Sequences encoding (G$_4$C$_2$)$_{73}$ repeats preceded by a start codon (ATG) were generated as previously described by primer hybridization (*Guo et al., 2018*). Similarly, the (G$_4$C$_2$)$_{73}$ construct was subcloned in place of the degenerated GA$_{65}$ sequence in the same NES/NLS-tagged GFP-containing vector. A GFP-only construct was generated by deletion of the GA from the same vector. NLS-LS has been previously described (*Frottin et al., 2019*). The plasmid encoding for S-mApple was generated from mApple-N1 (Addgene plasmid # 54567), a kind gift from Michael Davidson (*Shaner et al., 2008*). mApple was then cloned between BamHI and XbaI to replace GFP in a previously described plasmid encoding shuttle GFP (*Woerner et al., 2016*). c-myc-NES-β17, c-myc-NLS-β17, and Htt96Q plasmids have been previously described (*Woerner et al., 2016*). All relevant plasmid regions were verified by sequencing.

## Antibodies and dyes

The following primary antibodies were used in this study: nuclear pore complex proteins (Mab414), Abcam (24609); NPM1, Invitrogen (32-5200); NF-κB p65 (D14E12), Cell Signaling Technology (#8242); c-Myc-Cy3 (9E10), Sigma (C6594); amyloid fibrils, OC, Millipore (AB2286); GFP, Roche (11814460001); α-tubulin, Sigma Aldrich (T6199); RPA40, Santa Cruz (sc-374443); IκBα (L35A5), Cell Signaling Technology (#4814); phospho-NF-κB p65 (Ser536) (93H1), Cell Signaling Technology

(#3033); KPNA2, Abcam (ab70160); KPNA4, Abcam (ab84735); KPNB1, Abcam (ab2811); GAPDH, Millipore (MAB374); poly-PR, Proteintech (23979-1-AP); and poly-GR (5A2), Millipore (MABN778).

The following secondary antibodies were used: mouse IgG-Alexa488, Cell Signaling Technology (#4408); mouse IgG-Alexa555, Cell Signaling Technology (#4409); rabbit IgG-Alexa555, Cell Signaling Technology (#4413); mouse IgG-Alexa488, Invitrogen (21200); mouse IgG-Alexa 633, Invitrogen (A21053); anti-mouse IgG-Peroxidase, Sigma Aldrich (A4416); and anti-rabbit IgG-Peroxidase, Sigma Aldrich (A9169). The amyloid dye AmyTracker 680 (AmyT; Ebba Biotech AB) was used as previously described (*Frottin et al., 2019*). Briefly, cells were fixed in 4% paraformaldehyde in PBS (Gibco) for 20 min, washed with PBS, and permeabilized with Triton X-100 0.1% for 5 min. AmyT was used at 1:500 dilution and incubated with the samples for 1 hr at room temperature. Nuclei were counter-stained with 4',6-diamidine-2'-phenylindole dihydrochloride (DAPI, Molecular Probes).

## Immunofluorescence and image acquisition

Cells were grown on poly-L-lysine-coated coverslips (Neuvitro). Cells were fixed with 4% paraformal-dehyde, permeabilized with 0.1% Triton X-100, and blocked with 1% bovine serum albumin in PBS. Primary antibodies were applied in blocking buffer supplemented with 0.1% Triton X-100 and incubated overnight at 4°C. Appropriate fluorescent secondary antibodies at a dilution of 1:500 were applied for 60 min at room temperature. Nuclei were counterstained with DAPI before mounting samples with fluorescence-compatible mounting medium (DAKO).

Confocal microscopy was performed at MPIB Imaging Facility (Martinsried, Germany) on a ZEISS (Jena, Germany) LSM780 confocal laser scanning microscope equipped with a ZEISS Plan-APO 63×/NA1.46 oil immersion objective. In case of multi-fluorescence samples, a single-stained control sample was used to adjust emission and detection configuration to minimize spectral bleed-through. Images of cells with inclusions for co-localization studies were subjected to linear unmixing with spectra obtained from the single-stained samples using ZEN software. When fluorescence intensities were directly compared, acquisition settings and processing were kept identical. Images were analyzed with ImageJ (Rasband, W.S., National Institutes of Health, USA) and assembled in Adobe Photoshop CC (Adobe Systems Incorporated, Release 19.1.5).

## Cell viability assay

HEK293 cells were transfected by electroporation as previously described (*Woerner et al., 2016*). In brief, cells were electroporated with 20 µg of plasmid in 0.4 cm-gap electroporation cuvettes (Bio-Rad). Cells were electroporated at 225 V, $\infty$ Ω, 950 µF exponential wave in a GenePulser XCell System (Bio-Rad). After electroporation, cells were plated in a 24-well plate in triplicates. MTT assays were performed 3 days after transfection. The growth medium was replaced with fresh medium containing 5 µg/ml thiazolyl blue tetrazolium bromide (Sigma) for 1 hr. Formazan crystals were solubilized by addition of stop solution, containing 40% N,N-dimethylformamide (Sigma-Aldrich), 16% SDS (Sigma-Aldrich), and 2% (v/v) acetic acid (Sigma-Aldrich). Absorbance at 570 nm and 630 nm was then recorded. Alternatively, viability was measured using the CellTiter-Glo 2.0 Cell Viability Assay kit (Promega) in the same conditions.

## Determination of nuclear import/export with S-mApple

Cells were co-transfected with the indicated constructs and the reporter S-mApple. After 48 hr, cells were treated with 10 ng/ml of the CRM1 inhibitor Leptomycin B (LMB, Sigma Aldrich) in DMSO for 15 min before cell fixation. Control cells received DMSO. The relative concentration of S-mApple in the cytoplasm and nucleus was quantified by measuring the fluorescence intensity ratio in cells from three independent experiments. Fluorescence intensities were determined using ImageJ.

## Protein biosynthesis assay by click-chemistry

Protein biosynthesis assays were carried out using the click-it plus O-propargyl-puromycin (OPP) protein synthesis assay (Thermo Fisher Scientific) according to the manufacturer's instructions. Cells were transfected for 24 hr with the indicated construct before metabolic labeling and incubated with the OPP reagent for 30 min in normal growth conditions. As control, protein translation was inhibited with cycloheximide (CHX, Sigma-Aldrich) dissolved in PBS and applied at a final concentration of 1 mM. Samples were then fixed, permeabilized, and the click reaction performed as

recommended by the provider. Samples were subsequently analyzed by confocal microscopy. The concentration of labeled proteins was quantified by measuring the mean fluorescence intensity in 100–250 cells using ImageJ. A representative experiment of three independent experiments is shown.

### Fluorescence in situ hybridization (FISH)

Visualization of mRNA and $G_4C_2$ RNA by FISH was carried out as previously described (*Woerner et al., 2016*). HEK293 cells were fixed in 4% formaldehyde for 10 min at room temperature and permeabilized with 0.1% Triton X-100, both in UltraPure SCC buffer (Thermo Fisher Scientific). After washing with SSC buffer and with FISH buffer (10% formamide in SCC buffer), the probes (either T30 or $(C_4G_2)_5$) were hybridized in FISH buffer with 10% dextran sulphate (Sigma Aldrich) for 3 hr at 42°C, followed by additional washes in FISH buffer. When required, immunostaining was performed in PBS-based buffers afterward. The fraction of transfected cells displaying abnormal accumulation of mRNA (using poly-dT probe) within the nucleus was determined by confocal microscopy.

### Filter retardation assay

For the filter retardation assay (*Scherzinger et al., 1997*; *Wanker et al., 1999*), cells were harvested 24 hr after transfection with the indicated plasmids, lysed in radioimmunoprecipitation assay (RIPA) buffer (Thermo Fisher Scientific) and sonicated for 10 s. After incubation for 30 min, protein concentration was measured by Bradford assay (Bio-Rad), and equal amounts of lysates were filtered through a 0.2 µm pore size cellulose acetate membrane (GE Healthcare) and washed with lysis buffer. The membrane was subsequently immunoassayed with anti-GFP antibody. Antibody binding was detected using Luminata Forte Western HRP substrate (Millipore), and pictures were acquired with a LAS-3000 camera system (Fujifilm). AIDA (Raytest) software was used for analysis and quantitation.

### Statistics

Significance of differences between samples was determined using unpaired Student's t-test, unless stated otherwise. Significance levels: *p<0.05, **p<0.01, ***p<0.001.

## Acknowledgements

We thank R Klein, D Edbauer, R Körner, and R Sawarkar for helpful discussions. We acknowledge technical support by the MPIB Imaging facility. Funding: The research leading to these results has received funding from the European Commission under grant FP7 GA ERC-2012-SyG_318987–ToPAG. FF was supported by an EMBO Long Term Fellowship. MPB was supported by the Lehre@LMU Student Research Award Program of the Faculty of Biology at LMU Munich (sponsored by the Federal Ministry of Education and Research, funding no. 01PL17016).

## Additional information

### Competing interests

F Ulrich Hartl: Reviewing editor, *eLife*. The other authors declare that no competing interests exist.

### Funding

| Funder | Grant reference number | Author |
| --- | --- | --- |
| EMBO | Long term fellowship | Frédéric Frottin |
| European Commission | FP7 GA ERC-2012-SyG_318987-ToPAG | F Ulrich Hartl |
| Federal Ministry of Education and Research | 01PL17016 | Manuela Pérez-Berlanga |
| Max Planck Society | | Mark S Hipp |

The funders had no role in study design, data collection and interpretation, or the decision to submit the work for publication.

### Author contributions
Frédéric Frottin, Conceptualization, Supervision, Investigation, Writing - original draft, Writing - review and editing; Manuela Pérez-Berlanga, Investigation, Writing - review and editing; F Ulrich Hartl, Mark S Hipp, Conceptualization, Supervision, Writing - original draft, Writing - review and editing

### Author ORCIDs
Frédéric Frottin https://orcid.org/0000-0002-2756-7838
Manuela Pérez-Berlanga http://orcid.org/0000-0001-9064-9724
F Ulrich Hartl https://orcid.org/0000-0002-7941-135X
Mark S Hipp https://orcid.org/0000-0002-0497-3016

### Decision letter and Author response
Decision letter https://doi.org/10.7554/eLife.62718.sa1
Author response https://doi.org/10.7554/eLife.62718.sa2

## Additional files

### Supplementary files
• Transparent reporting form

### Data availability
All data generated or analysed during this study are included in the manuscript and supporting files.

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
