## [Decision Letter]

**Acceptance summary:**

A G4C2 repeat expansion in the C9orf72 gene causes amyotrophic lateral sclerosis and frontotemporal dementia. This work evaluates the relative contributions of dipeptide repeat proteins (DPR) and the repeat-containing mRNA derived from this repeat expansion in toxicity in a cellular model. Findings demonstrate that the G4C2 RNA is sufficient to reduce viability and inhibit nuclear transport in this model, and that the encoded DPR-containing proteins further contribute in an additive fashion to the toxic gain of function of this repeat expansion.

**Decision letter after peer review:**

Thank you for submitting your article "Multiple pathways of toxicity induced by C9orf72 dipeptide repeat aggregates and G4C2 RNA in a cellular model" for consideration by *eLife*. Your article has been reviewed by 3 peer reviewers, one of whom is a member of our Board of Reviewing Editors, and the evaluation has been overseen by David Ron as the Senior Editor. The reviewers have opted to remain anonymous.

The reviewers have discussed the reviews with one another and the Reviewing Editor has drafted this decision to help you prepare a revised submission.

The discovery in 2011 that a hexanucleotide (G4C2) expansion in the C9orf72 gene is the most frequent cause of inherited ALS and frontal temporal dementia has fuelled many reports testing whether slow, age-dependent dysfunction, degeneration, and ultimately death of selected classes of neurons is caused by G4C2 repeat containing RNAs (encoded by either sense or antisense DNA strands and accumulated into nuclear RNA foci) or by one or more of their five encoded dipeptide repeat polypeptides (DPRs) translated in an AUG independent manner. Multiple prior reports have reported evidence of toxicity of specific DPRs, several have argued for RNA-mediated toxicity, and several have argued for toxicity from both. Using synthetic genes that encode cytoplasmic or nuclear poly(65) GA aggregates in the absence of G4C2 hexanucleotide repeats, the authors demonstrate that the cellular compartment of polyGA aggregates is associated with specific aspects of cellular dysfunction (e.g., nuclear (but non-nucleolar) aggregates interfered with the protein quality control function of the nucleolus and cytoplasmic aggregates partially inhibited transport of proteins across the nuclear pore). However, expression of the G4C2 RNA, rather than DPRs, was associated with a decrease in cellular viability. Together these experiments demonstrate that DPRs and G4C2 RNA toxicity both impinge on cellular function/viability, likely in an additive fashion.

While all reviewers agreed that the quality of the studies is very high, there was extensive discussion about the direct relevance of these studies to ALS due to the single test system used – expression by transient transfection in HEK293T cells of a short (65 or 73) repeat. This system has very significant weaknesses that may preclude strong conclusions that go beyond the weaknesses of the prior work. These include: the cells are rapidly cycling; the level of expression relative to that in the neurons at risk in ALS or FTD is not determined; the repeats are far, far shorter than in the affected regions of the human nervous system where the repeat lengths are heterogeneous but in the many hundreds to many thousands; the properties of the short DPRs or RNAs are highly uncertain to reflect the levels and lengths of the corresponding DPRs or RNAs in human disease; there is no evidence that suggests that the lessons in these cycling, non-neuronal cells reflect the age-dependent selective damage within the neurons in human disease; and the viability assay does not uniquely distinguish inhibited cell cycling from cell death and is measured at 4 days, rather than the time scale of years that would be reflective of human disease.

While these weaknesses are also weaknesses of most of the preceding work, their limitations strongly reduce the novelty and success of the current work in setting it apart from the many prior reports. In addition, the reviewers felt that the previous studies on C9orf72 DPRs and/or RNA repeat toxicity were often superficially cited and citations of some recent publications were omitted. For further consideration, the authors are requested to re-examine the manuscript, in particular the introduction and discussion, to ensure that the reader can clearly understand the findings of previous studies so that the context is set for the current study. In addition, the authors are strongly urged to clearly define the relevance of their study in light of the points above.

Additional points for revision:

1. In Figure 1-supplement 1B, I do not see the peripheral staining with anti-amyloid antibody (OC) on the cytoplasmic inclusion. Is this image representative? If so, perhaps the authors can tone down this conclusion.

2. What is Beta23 in Figure 2D and Figure 2 supplement 1B?

3. In Figure 4-supplement 1C, the importin inclusions/sequestration of importin Beta-1 is not clear, particularly relative to the images in A and B? Perhaps the authors can comment on this in the text or show better images?

4. Representative images for protein translation experiments should be shown in Figure 5D for comparison to quantitated values.

---

## [Author Response]

The discovery in 2011 that a hexanucleotide (G4C2) expansion in the C9orf72 gene is the most frequent cause of inherited ALS and frontal temporal dementia has fuelled many reports testing whether slow, age-dependent dysfunction, degeneration, and ultimately death of selected classes of neurons is caused by G4C2 repeat containing RNAs (encoded by either sense or antisense DNA strands and accumulated into nuclear RNA foci) or by one or more of their five encoded dipeptide repeat polypeptides (DPRs) translated in an AUG independent manner. Multiple prior reports have reported evidence of toxicity of specific DPRs, several have argued for RNA-mediated toxicity, and several have argued for toxicity from both. Using synthetic genes that encode cytoplasmic or nuclear poly(65) GA aggregates in the absence of G4C2 hexanucleotide repeats, the authors demonstrate that the cellular compartment of polyGA aggregates is associated with specific aspects of cellular dysfunction (e.g., nuclear (but non-nucleolar) aggregates interfered with the protein quality control function of the nucleolus and cytoplasmic aggregates partially inhibited transport of proteins across the nuclear pore). However, expression of the G4C2 RNA, rather than DPRs, was associated with a decrease in cellular viability. Together these experiments demonstrate that DPRs and G4C2 RNA toxicity both impinge on cellular function/viability, likely in an additive fashion.While all reviewers agreed that the quality of the studies is very high, there was extensive discussion about the direct relevance of these studies to ALS due to the single test system used – expression by transient transfection in HEK293T cells of a short (65 or 73) repeat. This system has very significant weaknesses that may preclude strong conclusions that go beyond the weaknesses of the prior work. These include: the cells are rapidly cycling; the level of expression relative to that in the neurons at risk in ALS or FTD is not determined; the repeats are far, far shorter than in the affected regions of the human nervous system where the repeat lengths are heterogeneous but in the many hundreds to many thousands; the properties of the short DPRs or RNAs are highly uncertain to reflect the levels and lengths of the corresponding DPRs or RNAs in human disease; there is no evidence that suggests that the lessons in these cycling, non-neuronal cells reflect the age-dependent selective damage within the neurons in human disease; and the viability assay does not uniquely distinguish inhibited cell cycling from cell death and is measured at 4 days, rather than the time scale of years that would be reflective of human disease.While these weaknesses are also weaknesses of most of the preceding work, their limitations strongly reduce the novelty and success of the current work in setting it apart from the many prior reports. In addition, the reviewers felt that the previous studies on C9orf72 DPRs and/or RNA repeat toxicity were often superficially cited and citations of some recent publications were omitted. For further consideration, the authors are requested to re-examine the manuscript, in particular the introduction and discussion, to ensure that the reader can clearly understand the findings of previous studies so that the context is set for the current study. In addition, the authors are strongly urged to clearly define the relevance of their study in light of the points above.

We thank the reviewers for their helpful comments.

We have rewritten the introduction and the discussion and added a series of additional references to provide more context. We now state explicitly that we work in a cellular system (including in the title, the abstract, the introduction, the Results section and the discussion) and refer to the differences between the localization of arginine containing aggregates in cellular systems and patient material in the introduction and the discussion.

We would like to stress that our C9orf72 constructs recapitulate many features observed in patient samples, such as the formation of p62 positive cytoplasmic poly-GA inclusions and of nuclear G_4_C_2_-containing RNA foci. A critical advantage of the cellular system used is the fact that targeting poly-GA to the cytosol (NES-polyGA) does not result in overt toxicity. As a result, we can now (for the first time) clearly distinguish between G_4_C_2_ mRNA toxicity and proteotoxicity. We have emphasized this point in the revised manuscript.

However, we have also discussed the limitations of the cellular model system more extensively and state in the revised manuscript that “the longer G_4_C_2_ expansions, and the resulting DPRs, present in patients may have additional adverse effects that were not recapitulated with the constructs used here” , that “longer G_4_C_2_ sequences may mediate additional effects (West et al., 2020)” and that “we cannot exclude a DPR contribution to pathology in carriers of G_4_C_2_ expansions, where combinations of the various DPRs and mRNA are present simultaneously, and nondividing cells are exposed to these factors for extended time periods”. Notwithstanding these limitations, we demonstrate that the toxicity exerted by the repeat mRNA is substantially stronger when comparing constructs of the same length.

We hope that with these changes we have addressed the general concerns raised by the reviewers.

Additional points for revision:1. In Figure 1-supplement 1B, I do not see the peripheral staining with anti-amyloid antibody (OC) on the cytoplasmic inclusion. Is this image representative? If so, perhaps the authors can tone down this conclusion.

We have rephrased this statement and exchanged the image with an example where the peripheral staining is better visible.

2. What is Beta23 in Figure 2D and Figure 2 supplement 1B?

Beta23 is an aggregation-prone beta-sheet protein. We are using it to control for compartment specific toxic effects of protein aggregation. This is now explained and appropriate references have been added (Woerner et al. 2016, and West et al. 1999)

3. In Figure 4-supplement 1C, the importin inclusions/sequestration of importin Beta-1 is not clear, particularly relative to the images in A and B? Perhaps the authors can comment on this in the text or show better images?

We have altered the text to emphasize that the effect for KPNA2 and KPNA4 is stronger than for KPNB1, and have added fluorescence profiles to Figure 4-supp1C to show the elevated levels of the importins more clearly.

4. Representative images for protein translation experiments should be shown in Figure 5D for comparison to quantitated values.

We have now moved the examples for the quantification of the experiment from Supplementary figure 5D to the main Figure 5.